# Loss of Otopetrin 1 affects thermoregulation during fasting in mice

**Yu-Hsiang Tu[1], Naili Liu[2], Cuiying Xiao[1], Oksana Gavrilova[2], Marc L. Reitman**[1]*

**1** Diabetes, Endocrinology and Obesity Branch, National Institute of Diabetes and Digestive and Kidney Diseases, National Institutes of Health, Bethesda, MD, United States of America, **2** Mouse Metabolism Core, National Institute of Diabetes and Digestive and Kidney Diseases, National Institutes of Health, Bethesda, MD, United States of America

* marc.reitman@nih.gov

**Data Availability Statement:** All relevant data are within the paper and its Supporting Information files.

**Funding:** This research was supported by the Intramural Research Program of the National

## Abstract

### Objective

Otopetrin 1 (OTOP1) is a proton channel that is highly expressed in brown adipose tissue. We examined the physiology of *Otop1*⁻ᐟ⁻ mice, which lack functional OTOP1.

### Methods

Mice were studied by indirect calorimetry and telemetric ambulatory body temperature monitoring. Mitochondrial function was measured as oxygen consumption and extracellular acidification.

### Results

*Otop1*⁻ᐟ⁻ mice had similar body temperatures as control mice at baseline and in response to cold and hot ambient temperatures. However, in response to fasting the *Otop1*⁻ᐟ⁻ mice exhibited an exaggerated hypothermia and hypometabolism. Similarly, in ex vivo tests of *Otop1*⁻ᐟ⁻ brown adipose tissue mitochondrial function, there was no change in baseline oxygen consumption, but the oxygen consumption was reduced after maximal uncoupling with FCCP and increased upon stimulation with the $\beta_3$-adrenergic agonist CL316243. Mast cells also express *Otop1*, and *Otop1*⁻ᐟ⁻ mice had intact, possibly greater hypothermia in response to mast cell activation by the adenosine $A_3$ receptor agonist MRS5698. No increase in insulin resistance was observed in the *Otop1*⁻ᐟ⁻ mice.

### Conclusions

Loss of OTOP1 does not change basal function of brown adipose tissue but affects stimulated responses.

Institutes of Health, National Institute of Diabetes and Digestive and Kidney Diseases (MLR, ZIA DK075062; MLR, ZIA DK075064;OG ZIA DK070002). The funders had no role in study design, data collection and analysis, decision to publish, or preparation of the manuscript.

**Competing interests:** The authors have declared that no competing interests exist.

**Abbreviations:** BAT, brown adipose tissue; RER, respiratory exchange ratio; $T_a$, ambient temperature(s); $T_b$, core body temperature; TEE, total energy expenditure; WAT, white adipose tissue.

# 1 Introduction

Otopetrins are proton-selective channels [1] with a dimeric multi-transmembrane structure [2]. *Otop1* is the gene altered by the mouse *tlt* mutation [3]. The phenotype of *Otop1$^{tlt/tlt}$* mice, which have a homozygous A151E mutation in otopetrin 1, includes a tilted head and the inability to orient properly while swimming. These features are attributed to abnormal otoconial morphogenesis, which causes impaired positional sensing, with a similar phenotype in mutant zebrafish [4]. A clue to *Otop1*'s molecular function was that it affects intracellular calcium regulation by the P2Y purinergic receptor in cell lines [5] and in vestibular supporting cells [6]. More recently, the proton channel function of otopetrin 1 was discovered and the proton-carrying ability was identified as crucial for sour taste perception [1,7,8].

Multiple paralogs have been identified. For example, *OtopLA* is required for acid taste in *Drosophila* [9], and *Otop2l* is needed for biomineralization in sea urchin larvae [10]. The otopetrin family includes two other members in mice and humans, *Otop2* and *Otop3*. *Otop1* is expressed in brown adipose tissue (BAT), mast cells, and adrenal gland, in addition to inner ear and sour taste cells, while *Otop2* is found in the olfactory bulb and stomach, and *Otop3* is in stomach and small intestine [1].

The role of *Otop1* in BAT has not been explored. BAT's function is to generate heat to maintain body temperature ($T_b$) [11]. The main mechanism of BAT heat generation is a futile cycle where uncoupling protein 1 (UCP1) 'leaks' protons across the inner mitochondrial membrane, diverting the electrochemical gradient to generate heat instead of ATP [12]. *Otop1$^{tlt/tlt}$* mice have been reported to have increased insulin resistance, hepatic steatosis, and adipose tissue inflammation when fed a high-fat diet [13], a phenotype consistent with abnormal BAT function. Here we investigated the function of OTOP1 in BAT, examining the thermal and metabolic phenotypes of null cells and mice.

# 2 Methods

## 2.1 Animals

Mice were obtained as follows: C57BL/6J (Jax# 000664) and *Ucp1$^{-/-}$* (B6.129-*Ucp1$^{tm1Kz}$*/J [14]). *Otop1$^{-/-}$* (*Otop1$^{em1Lmn}$* [7], from Dr. Emily Liman) mice have no detectable proton current in sour taste cells, in contrast to *Otop1$^{tlt/tlt}$* mice which have a small residual proton current in sour taste cells [1]. *Otop1$^{-/-}$*;*Ucp1$^{-/-}$* mice were generated by breeding the *Otop1$^{-/-}$* and *Ucp1$^{-/-}$* mice. *Otop1$^{-/-}$* mice were bred with C57BL/6J mice and littermate mice were used as controls. Mice from multiple litters were used for all studies. Mice were singly housed after E-Mitter implantation surgery at 23˚C with lights on 6 am–6 pm. Chow (7022, Envigo, Indianapolis, IN) or a 60% high fat diet (D12492, Research Diets, New Brunswick, NJ), and water were available ad libitum, with exception of fasting experiments as indicated. Mice received daily visual health status checks according to NIH guidelines. Animal studies were approved by the NIDDK/NIH Animal Care and Use Committee, protocol K016-DEOB-20.

## 2.2 Analysis of BAT mitochondrial function

Seahorse analysis of tissue fragments was modified from the method of [15]. We studied freshly isolated BAT fragments, rather than cultured adipocytes, to avoid any possible effects of cell culture. Ad libitum, chow-fed mice (at 8–12 weeks) were euthanized using $CO_2$ at noon and interscapular brown fat was isolated, minced, and very lightly digested (Collagenase B; Roche 11088831001) 1.5 mg/mL (1 mL/BAT lobe), NaCl 125 mM, KCl 5 mM, $CaCl_2$ 1.3 mM, glucose 5 mM, penicillin 100U/mL, streptomycin 100μg/mL, bovine serum albumin [BSA] 4%) at 37 ˚C for 0.5–1 h with shaking (200 rpm). The digested tissue was then gravity filtered

through a 100 μm filter (pluriSelect 43-50100-51). The filter-retained tissue, consisting visually of small clumps of brown adipose tissue, was washed with DMEM-F12 and triturated 10–12 times using a P1000 pipetman using wide bore 'genomic' tips until the tissue suspension passed freely; if clumps did not aspirate, they were discarded. Next, 20 μl of suspension was added onto the mesh of XFe24 Islet Capture microplate, which was placed upside-down (since the tissue clumps float) in a 24-well microplate (Agilent 103518–100). The microplate was rinsed once with Seahorse medium (Agilent 103575–100) and with Seahorse medium with nutrient supplements (Seahorse DMEM medium, HEPES 5 mM, glucose 10 mM, pyruvate 1 mM, glutamine 2 mM, carnitine 0.5 mM, fatty acid free BSA 33.3 μM (Roche 03117057001), with or without palmitate 200 μM (Sodium palmitate; Sigma P9767) in 150 mM NaCl, along with 33.3 μM fatty acid free BSA). Plates were centrifuged (300g, room temperture, 3 min) to remove any bubbles, incubated (37 ˚C, 1 hour), the medium was changed, and plates analyzed with the Seahorse XFe24 Analyzer. The samples were treated with CL316243 (10 μM final; Sigma C5976) or vehicle (Seahorse medium with nutrients), oligomycin (15 μM final; Sigma O4876), FCCP (8 μM final; Sigma C2920), rotenone (5 μM final; Sigma 45656) plus antimycin A (10 μM final; Sigma A8674), as indicated.

## 2.3 Indirect calorimetry and core body temperature measurement

Littermate wild type and *Otop1*[-/-] mice were implanted intraperitoneally with G2 E-Mitter transponders (Starr Life Sciences, Oakmont, PA) and their $T_b$, total energy expenditure (TEE), respiratory exchange ratio (RER), and physical activity by beam break were measured using an Oxymax/CLAMS system (Columbus Instruments), as described [16]. The measurements were performed under different conditions, including ambient temperatures of 8 ˚C, 23 ˚C, 35 ˚C, fasting at 23 ˚C, and CL316243 (0.1 mg/kg, i.p.; $ED_{50}$ is 0.004–0.008 mg/kg [17]) at 30 ˚C.

## 2.4 Light and electron microscopy

For light microscopy, interscapular BAT from WT and *Otop1*[-/-] male chow-fed mice was harvested, fixed with paraformaldehyde, sectioned and stained with hematoxylin and eosin (American Histo Labs Inc, Gaithersburg, MD). Images were captured using an Olympus BX61 motorized microscope with Olympus BX-UCB hardware (VS120 slide scanner) and processed using Olympus OlyVIA software. Images were minimally processed to adjust brightness and contrast.

For transmission electron microscopy, interscapular BAT from chow-fed WT and *Otop1*[-/-] male mice was dissected, fixed (2.5% Glutaraldehyde, 1% Paraformaldehyde in 120 mM sodium cacodylate buffer, pH 7.4), and studied at the NHLBI Electron Microscopy Core Facility (NHLBI, NIH).

## 2.5 Gene expression and DNA levels

For RNA seq, interscapular BAT from ad libitum chow-fed WT and *Otop1*[-/-] male mice was dissected and frozen on dry ice. RNA isolation, library preparation, sequencing, and sequence alignment were performed by Quick Biology (Pasadena, CA). The aligned sequences were displayed using the IGV browser (https://software.broadinstitute.org/software/igv/).

For RT-PCR, BAT and inguinal and epididimal white adipose tissue (WAT) from WT and *Otop1*[-/-] male ad libitum chow/HFD-fed mice were dissected at noon. RNA extracted (RNeasy Plus kit, Qiagen), and reverse transcribed (Transcriptor First strand cDNA synthesis kit, Roche). cDNA was quantitated by PCR (primers in **S1 Table**) with SYBR Green PCR Master Mix (ThermoFisher) using QuantStudio 6 Flex (Applied Biosystem) and normalized to each sample's *Tbp* levels.

To measure relative mitochondrial content, the mitochondrial to nuclear DNA ratio was measured by qPCR of the *Nd1* (mitochondrial) and *Hk2* (nuclear) genes. DNA samples were collected from BAT of male chow-fed mice, aged 8–12 weeks. Primers: *Nd1* (Fwd, `CTAGCAGAAACAAACCGGGC`; Rev, `CCGGCTGCGTATTCTACGTT`); *Hk2* (Fwd, `GCCAGCCTCTCCTGATTTTAGTGT`; Rev, `GGGAACACAAAAGACCTCTTCTGG`) [18].

## 2.6 Western blot analysis

BAT lysates from WT and *Otop1⁻/⁻* mice were prepared, separated using reducing SDS PAGE gels (Bolt Bis-Tris Plus mini-gel; ThermoFisher Scientific NW0412A), and transferred to Immobilon®-P PVDF Membrane (Millipore IPVH304F0). Primary antibodies targeting OTOP1 (1:500 dilution, Novus NBP1-86306; 1:500, Alomone AHC-005; 1:500, Abnova H00133060-M02; 1:500, MyBioSource MBS6003792), GAPDH (1:5000, Sigma G8795), UCP1 (1:1000, Sigma U6382), and α-tubulin (1:5000; Sigma T6074) were used. HRP-conjugated secondary antibodies (1:5000, Goat anti-Rabbit IgG, Thermofisher Scientific A16116; 1:5000, Goat anti-Mouse IgG, ThermoFisher Scientific 31430) were used with SuperSignal™ West Pico PLUS Chemiluminescent Substrate as recommended (ThermoFisher Scientific 34580).

## 2.7 Glucose and insulin tolerance tests and metabolite profiles

Assays were performed as described [19]. In brief, glucose (2 g/kg, i.p., for chow-fed mice and 1 g/kg, i.p., for mice on an HFD) tolerance tests were performed following an overnight fast, with AUC calculated from the baseline. Insulin (0.75 unit/kg, i.p.) tolerance tests were performed at 3 pm in nonfasted mice. Blood was collected at 11 am by tail bleeding at 25 weeks of age for measurements of ad libitum fed blood glucose and serum insulin, triglycerides, free fatty acids, cholesterol, β-hydroxybutyrate, and leptin levels. Glucose was measured with a Glucometer Contour (Bayer, Mishawaka, IN). Free fatty acids (Fujifilm Waco Diagnostics, Mountain View, CA, reagents 999–34691, 995–34791, 991–34891, 993–35191), triglycerides (Pointe Scientific Inc., Canton, MI, T7532-120), cholesterol (Thermo Scientific, Middletown, VA, TR13421), and β-hydroxybutyrate (BioVision, Milpitas, CA, K651-100) were measured using the indicated colorimetric assays. Leptin (R&D Systems, Minneapolis, MN, MOB00) and insulin (Crystal Chem, Downers Grove, IL, 90010 using mouse insulin standard 90020) were measured by ELISA.

## 2.8 Statistical analysis

Statistical analyses were done using Prism (version 9.0; GraphPad Software, Inc.).

## 3 Results

### 3.1 Characteristics of *Otop1⁻/⁻* mice

*Otop1* is highly expressed in mouse BAT [20,21]. We confirmed this by RT-PCR, with BAT *Otop1* mRNA levels 6.3- or 46.2-fold higher than in inguinal or epididymal WAT, respectively (**Fig 1A**). Using RNA-seq, BAT *Otop1* mRNA transcript reads were abundant at 303 reads per million in wild type and reduced by 35% in the *Otop1⁻/⁻* mice (**Fig 1B**). The *Otop1⁻/⁻* mRNAs carried the expected 38-bp deletion starting at amino acid 40, which disrupts the reading frame [7] (**S1A Fig**).

The *Otop1⁻/⁻* mice are null for OTOP1 function [7] and exhibit a head tilt as seen in *Otop1ᵗˡᵗ/ᵗˡᵗ* mice [3], but no other obvious abnormalities or increased lethality were observed. We were unsuccessful in measuring OTOP1 protein levels by Western blotting of BAT since

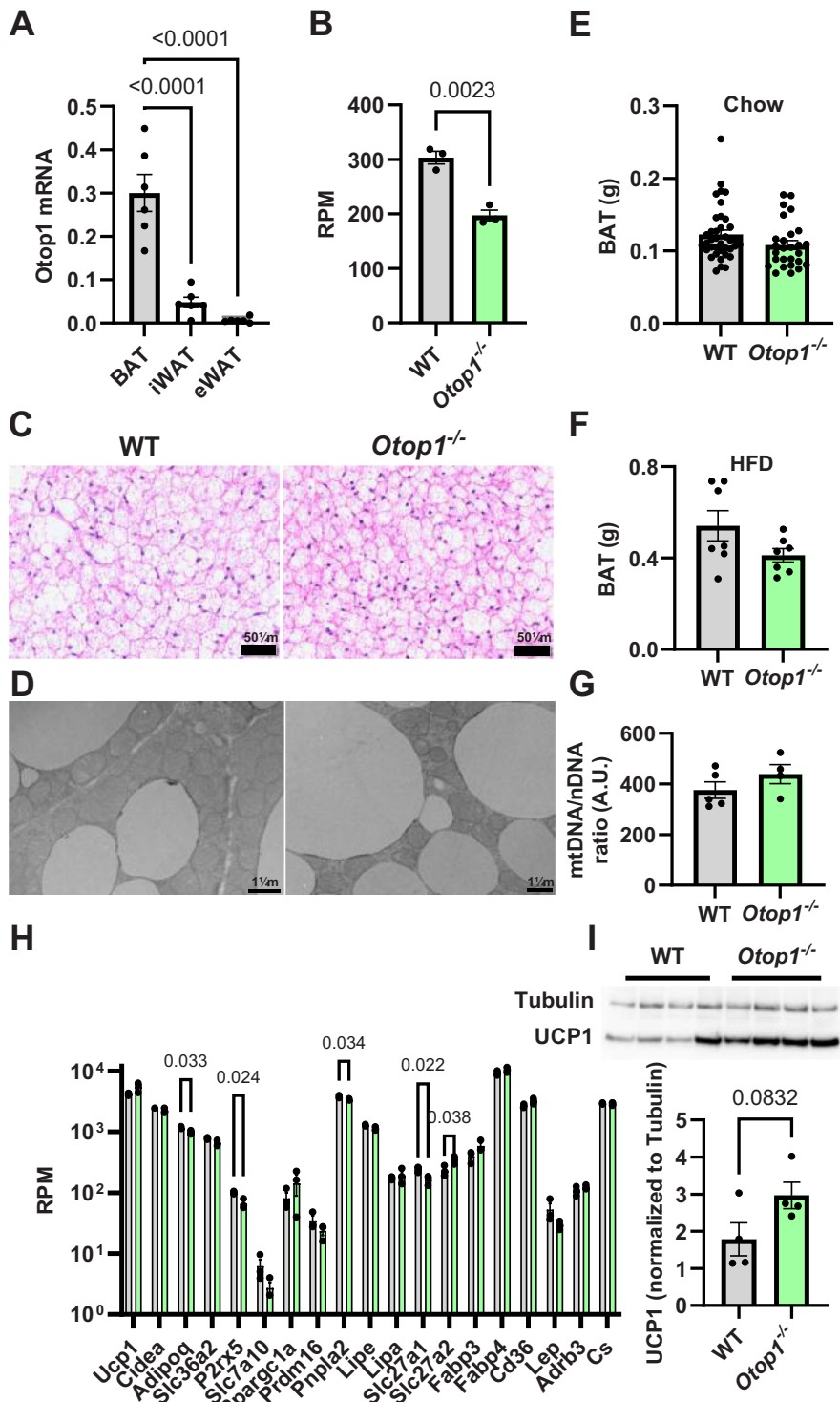

**Fig 1. Characteristics of *Otop1*<sup>-/-</sup> mice.** (A) *Otop1* expression in brown adipose tissue (BAT), inguinal, and epididymal white adipose tissue (iWAT, eWAT) of wild type mice measured by qRT-PCR, normalized to *Tbp* (n = 6/ group; p-value from 1-way ANOVA with Tukey post-hoc testing). (B) BAT *Otop1* expression measured by RNA-seq in WT and *Otop1*<sup>-/-</sup> mice (n = 3/group; p-value from unpaired t-test). RPM, reads per million. (C) Representative light microscopy (hematoxylin and eosin stain) and (D) electron microscopy images of BAT from male, chow-fed WT and *Otop1*<sup>-/-</sup> mice, age 11 weeks. (E) BAT weight of chow-fed male mice, age 8–12 weeks; n = 38 or 27/group. (F) BAT weight of HFD-fed male mice, age 45 weeks; n = 7/group. (G) BAT mitochondrial content. The mitochondrial:nuclear

DNA ratio was measured in chow-fed male mice 8–12 weeks old, n = 4-5/group as the ratio of *Nd1*:*Hk2* qPCR products. A.U., arbitrary units. (H) Gene expression level of known BAT genes in chow-fed male WT and *Otop1*$^{-/-}$ mice (aged 13–16 weeks), measured by RNA-Seq. n = 3/group; p-value from unpaired t-test without multiplicity correction, (I) UCP1 Western blot. UCP1 protein level is normalized to α-tubulin at the same lane. n = 4/group; p-value from unpaired t-test.

none of the four commercial antibodies putatively to OTOP1 detected a band of the correct size in wild type mice (see Methods).

BAT appearance by light and electron microscopy, BAT weight, and BAT mitochondrial content of *Otop1*$^{-/-}$ mice were all indistinguishable from wild type controls (**Fig 1C–1G**). *Ucp1* levels were unchanged in *Otop1*$^{-/-}$ mice at the mRNA (p = 0.14; **Fig 1H**) and protein (p = 0.083; **Fig 1I**) levels.

To screen for effects of *Otop1* loss, BAT mRNA expression profiles were examined. Using RNA-Seq, the mRNAs most different from controls encoded histocompatibility and heat shock proteins (**S2A Fig, S2 Table**). However, these did not replicate in an independent experiment using RT-PCR (**S2B Fig**). Analysis of BAT marker genes did not find large differences from controls, with *Slc27a2* (FATP2) possibly being an exception, showing a 53% increase (**Fig 1H**) and 3.5-fold increase in a replication qPCR experiment (**S2C Fig**), although this increase was not present after 8-weeks on a HFD (**S2D Fig**).

On a chow diet *Otop1*$^{-/-}$ male and female mice had similar body weights as controls (**Fig 2A and 2B**), also seen in two independent male cohorts (e.g., 28.0 g vs 31.7 g, wild type vs *Otop1*$^{-/-}$ at age 19 weeks, p = 0.02; and 36.2 g vs 33.1 g, wild type vs *Otop1*$^{-/-}$ at age 38 weeks, p = 0.16; n = 5-7/group). No difference was elicited by a high fat diet (HFD) to induce obesity (**Fig 2A and 2B**). During 8 weeks on an HFD, both male and female *Otop1*$^{-/-}$ mice showed no significant difference in food intake or feed efficiency (**Fig 2C–2F**). In summary, aside from the head tilt, we identified no abnormal physical characteristics in *Otop1*$^{-/-}$ mice.

## 3.2 Effect of *Otop1* loss on mitochondrial function

To examine the loss of *Otop1* on mitochondrial function, we studied freshly isolated BAT fragments [15]. The wild type BAT had the expected properties by Seahorse analysis, including a minimal effect of oligomycin indicating little ATP-linked respiration, a robust FCCP-induced maximally uncoupled oxygen consumption rate (OCR) of 210% of baseline, and a non-mitochondrial (antimycin- and rotenone-inhibited) OCR of ~30% of baseline (**Fig 3A–3C**), all of which are comparable to studies of BAT adipocytes and improved over other tissue fragment assays [15,22]. These experiments included added palmitate in the medium. Loss of *Otop1* did not affect the basal OCR, ATP-linked respiration, or non-mitochondrial respiration, but significantly reduced maximal mitochondrial respiration to 154% of baseline (**Fig 3A–3C**).

When palmitate was not added to the medium, the basal OCR was also similar in control and *Otop1*$^{-/-}$ mice (**Fig 3D and 3E**). Interestingly, the FCCP:basal OCR ratio, which was lower in *Otop1*$^{-/-}$ mice when palmitate was added, was similar to WT when palmitate was not added (**Fig 3F**).

We next tested for interaction between OTOP1 and UCP1 function, reasoning that *Ucp1*$^{-/-}$ mice have deficient BAT function [14] which might unmask a subtle *Otop1*$^{-/-}$ phenotype. With added palmitate in the medium, loss of *Ucp1* by itself greatly reduced basal BAT OCR (114 vs 390 pmol/min/μg DNA, *Ucp1*$^{-/-}$ vs WT, as expected [23]) but the fold increase with FCCP was not significantly affected (217% in *Ucp1*$^{-/-}$ vs 210% in WT). In *Otop1*$^{-/-}$;*Ucp1*$^{-/-}$ (double knockout, DKO) mice there was no significant difference in basal and FCCP-induced OCR compared to *Ucp1*$^{-/-}$ mice (**Fig 3A–3C**). There was no effect of *Otop1* or *Ucp1* genotype on ATP-

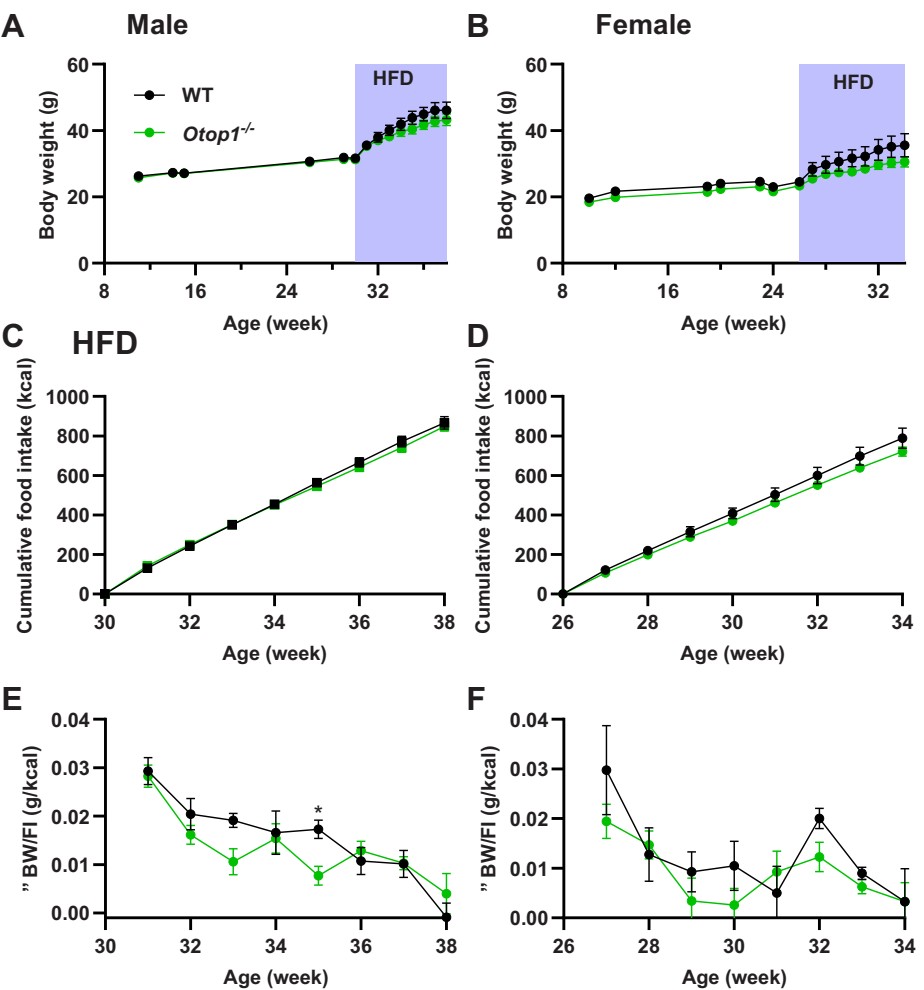

**Fig 2. *Otop1*<sup></sup> mice have similar body weight and food intake as WT mice.** Body weight of littermate (A) male and (B) female WT and *Otop1*<sup></sup> mice. Diet was switched from chow to high fat diet (HFD) at 30 (male) or 26 (female) weeks of age, n = 4-7/group. Cumulative HFD food intake for the (C) male and (D) female mice. Weekly feed efficiency (change in body weight/food intake) during HFD in (E) male and (F) female mice.

linked or on non-mitochondrial respiration. These data show that *Otop1* loss reduces maximal, but not basal respiration, in contrast to *Ucp1* loss, which reduces basal respiration. No interaction was detected between the *Otop1*-null and *Ucp1*-null genotypes.

## 3.3 Thermal physiology of *Otop1*⁻/⁻ mice

We next examined the thermal physiology of *Otop1*⁻/⁻ mice as a *in vivo* screen of BAT function. *Otop1*⁻/⁻ male mice showed a normal diurnal rhythm in $T_b$, TEE, RER, and food intake (**Fig 4A–4J**). At $T_a$s of 8 ˚C, 23 ˚C, and 35 ˚C, the *Otop1*⁻/⁻ and control mice had similar $T_b$ and TEE, and when compared to 23 ˚C both showed the expected TEE increase at 8 ˚C and TEE reduction and $T_b$ increase at 35 ˚C. These results rule out major defects in BAT function. *Otop1*⁻/⁻ mice ate similar amounts per 24 h as controls, but with a greater percentage during the dark phase (71% in WT vs 79% in *Otop1*⁻/⁻, p = 0.045). Physical activity was reduced in the *Otop1*⁻/⁻ mice at 23 ˚C, likely related to their vestibular deficit, since another mouse with a vestibular deficit also has decreased physical activity [24].

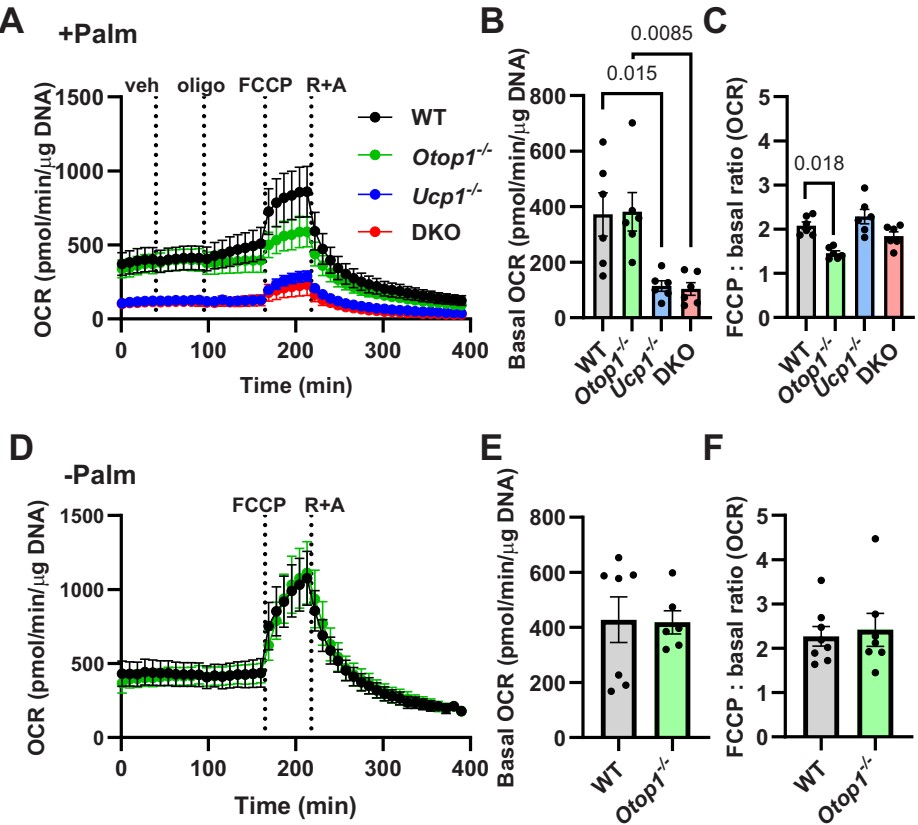

**Fig 3. Loss of *Otop1* decreases BAT maximal mitochondrial respiration in palmitate-rich environment.** (A) BAT oxygen consumption rate (OCR) in WT, *Otop1⁻/⁻*, *Ucp1⁻/⁻*, and DKO (*Otop1⁻/⁻;Ucp1⁻/⁻*) male mice, with palmitate added in the medium. Vehicle (veh), oligomycin (oligo), FCCP, and rotenone + antimycin (R+A) were added at the indicated times. (B) Basal OCR, and (C) FCCP:basal OCR ratio (n = 6/group, p-values by 2-way ANOVA with Tukey post-hoc testing. (D-F) BAT OCR without additional palmitate in the medium in WT and *Otop1⁻/⁻* mice, otherwise as in (A-C), n = 6-8/group.

During a 24-h fast at a $T_a$ of 23 ˚C, the *Otop1⁻/⁻* mice showed a greater reduction in $T_b$, TEE, and physical activity and a more rapid reduction in RER than wild type controls (**Fig 4K–4N**). In addition, after a 24-hour fast, mRNA levels of some BAT genes, including *Cidea*, *Pnpla2* (ATGL), and *Slc27a2* (FATP2) were higher in the *Otop1⁻/⁻* mice (**S3 Fig**).

In contrast to chow-fed mice, when male DIO mice were fasted, they did not have an augmented decrease in $T_b$ and activity, likely because fasting is not as energetically challenging in obese mice. The baseline $T_b$ of DIO *Otop1⁻/⁻* and WT mice were also not different (**Fig 4O–4R**).

We next studied female *Otop1⁻/⁻* mice, which replicated the results seen in males, with a similar diurnal rhythm of $T_b$, TEE, RER, and food intake and reduced dark phase physical activity compared to wild type mice (**Fig 5A–5J**). Upon fasting, the chow-fed female *Otop1⁻/⁻* mice, like the males, showed an exaggerated response, with a lower $T_b$, TEE, and physical activity than the WT mice (**Fig 5K–5N**). The effect of fasting on female DIO *Otop1⁻/⁻* mice was variable and not significantly different from wild type mice (**Fig 5O–5R**).

In summary, *Otop1⁻/⁻* mice respond like wild type mice to changes in ambient temperature, sometimes also with decreased physical activity. *Otop1⁻/⁻* mice have an amplified response to fasting in both sexes, which is lost when mice have greater adiposity from an HFD.

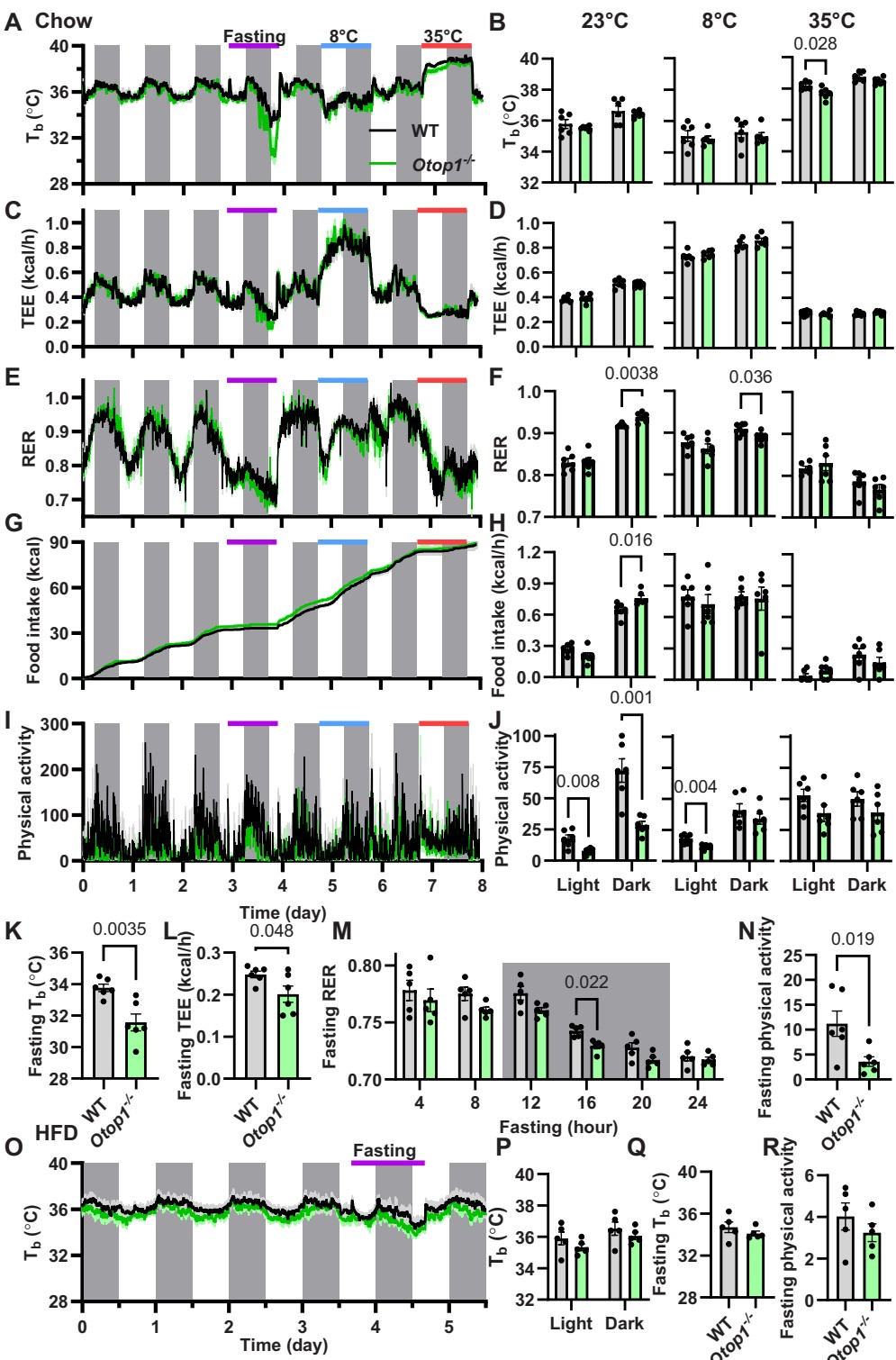

**Fig 4. *Otop1⁻/⁻* male mice have an augmented response to fasting.** (A-J) Male 14-week-old wild type (WT) and *Otop1⁻/⁻* mice on a chow diet at 23 ˚C were exposed to fasting (day 3), 8 ˚C (day 5), or 35 ˚C (day 7) while (A) $T_b$, (C) TEE, (E) RER (respiratory exchange ratio), (G) cumulative food intake, and (I) physical activity were measured. Mean (B) $T_b$, (D) TEE, (F) RER, (H) hourly food intake, and (J) physical activity during the light and dark periods at the indicated ambient temperatures. (K) Mean $T_b$, (L) TEE, and (N) physical activity, during the last 4 hours of fasting (n = 6/group, p-value calculated by unpaired t-test). (M) RER in 4-hour intervals during fasting (n = 6/group, 2-way

ANOVA with Sidak post-hoc testing). (O-R) Male 39-week-old mice after 8 weeks on a high fat diet (HFD), body weight 46.2 ± 2.8 g in WT and 45.0 ± 1.7 g in $Otop1^{-/-}$. (O) $T_b$, (P) mean $T_b$ of days 1–3, and (Q,R) fasting $T_b$ and physical activity during the last 4 hours of fasting (n = 5/group, p-values by unpaired t-test).

## 3.4 $Otop1^{-/-}$ mice have an enhanced response to a $\beta_3$-adrenergic agonist

We used a selective $\beta_3$-adrenergic agonist, CL316243, to measure stimulated BAT function. *Ex vivo*, CL316243 increased OCR more in $Otop1^{-/-}$ than in WT BAT. When no exogenous palmitate was added, the increase was to 161% in WT vs 270% in $Otop1^{-/-}$ (p<0.0001), while with added palmitate, there was no significant difference (**Fig 6A and 6B**). We studied the bioenergetic phenotype (aerobic-glycolytic balance) by comparing the OCR with the extracellular acidification rate (ECAR) in the Seahorse assay [25–27]. The basal OCR/ECAR ratio was similar in WT and $Otop1^{-/-}$ BAT, with an increase in OCR/ECAR ratio when palmitate was added (**Fig 6C**). Interestingly, without added palmitate CL316243 stimulation increased the OCR/ECAR ratio more in $Otop1^{-/-}$ BAT, reaching the level seen with added palmitate.

*In vivo*, CL316243 treatment of mice at 30 ˚C increased $T_b$ similarly in male WT and $Otop1^{-/-}$ mice (**Fig 6D**). CL316243 treatment produced a greater TEE increase in the $Otop1^{-/-}$ mice (**Fig 6F**), while it decreased RER (**Fig 6H**) similarly in $Otop1^{-/-}$ and control mice. In female $Otop1^{-/-}$ mice, the responses to CL316243 were similar to those in males (**Fig 6E, 6G and 6I**), but the TEE increase was not significantly greater in $Otop1^{-/-}$ females (217% in $Otop1^{-/-}$ vs 194% in WT, p = 0.29) (**Fig 6G**).

We also looked for interaction between the $Otop1$ and $Ucp1$ genotypes in the *ex vivo* response to treatment with a $\beta_3$-adrenergic agonist in a palmitate-rich environment. $Ucp1^{-/-}$ BAT showed a smaller increase in OCR after CL316243 treatment (123% of baseline in $Ucp1^{-/-}$ vs 170% in WT, p = 0.002). The increase was similar in BAT from $Ucp1^{-/-}$ and DKO mice (118%) and these genotypes had a significantly lower OCR/ECAR ratio both at baseline and after CL316243 treatment (**S4 Fig**). This suggests that the BAT energy phenotype shifted towards glycolysis with the loss of UCP1. *In vivo*, the $Ucp1^{-/-}$ genotype, either by itself or with $Otop1^{-/-}$ in the DKO mice, caused a severely reduced $T_b$ and TEE response to CL316243, while RER was similar (**S5 Fig**).

These results demonstrate that $Otop1^{-/-}$ mice show increased energy expenditure in response to $\beta_3$-adrenergic BAT stimulation, both *ex vivo* and *in vivo*, and adding palmitate abolished this effect *ex vivo*.

## 3.5 No major glucose or lipid phenotype in $Otop1^{-/-}$ mice

A previous study reported WAT inflammation and insulin resistance in HFD-fed $Otop1^{tlt/tlt}$ mice [13]. We next studied insulin resistance in $Otop1^{-/-}$ mice. On both a chow diet and after 8 weeks on an HFD, male WT and $Otop1^{-/-}$ mice had similar serum levels of fatty acids, cholesterol, $\beta$-hydroxybutyrate, and leptin, with lower triglyceride levels in the HFD $Otop1^{-/-}$ group (**Fig 7A–7E**). The fed and fasted glucose and insulin levels, glucose tolerance tests, and insulin sensitivity as measured by insulin tolerance tests were also similar (**Fig 7F–7N**). In female mice, there were also no differences in these parameters except for an improved GTT on the HFD, suggesting no major sex differences (**S6 Fig**).

WAT RNA levels of metabolic and inflammatory genes were measured in HFD-fed $Otop1^{-/-}$ mice. Relative to control mice, there were reductions in lipid metabolism genes (*Lipe* (HSL), *Fasn*; and non-significantly in *Pnpla2* (ATGL), *Ehhadh*) in epididymal WAT, but not in inguinal WAT (**Fig 7O and 7P**). There were no changes in mRNA markers of inflammation (*Oasl1*, *Oasl2*, *Ifi44*, *Ifitm3*, *Ifih1*, *Ccl2*, *Ifng*, *Tnf*) or of macrophages (*Emr2* (F4/80), *Lgals3*

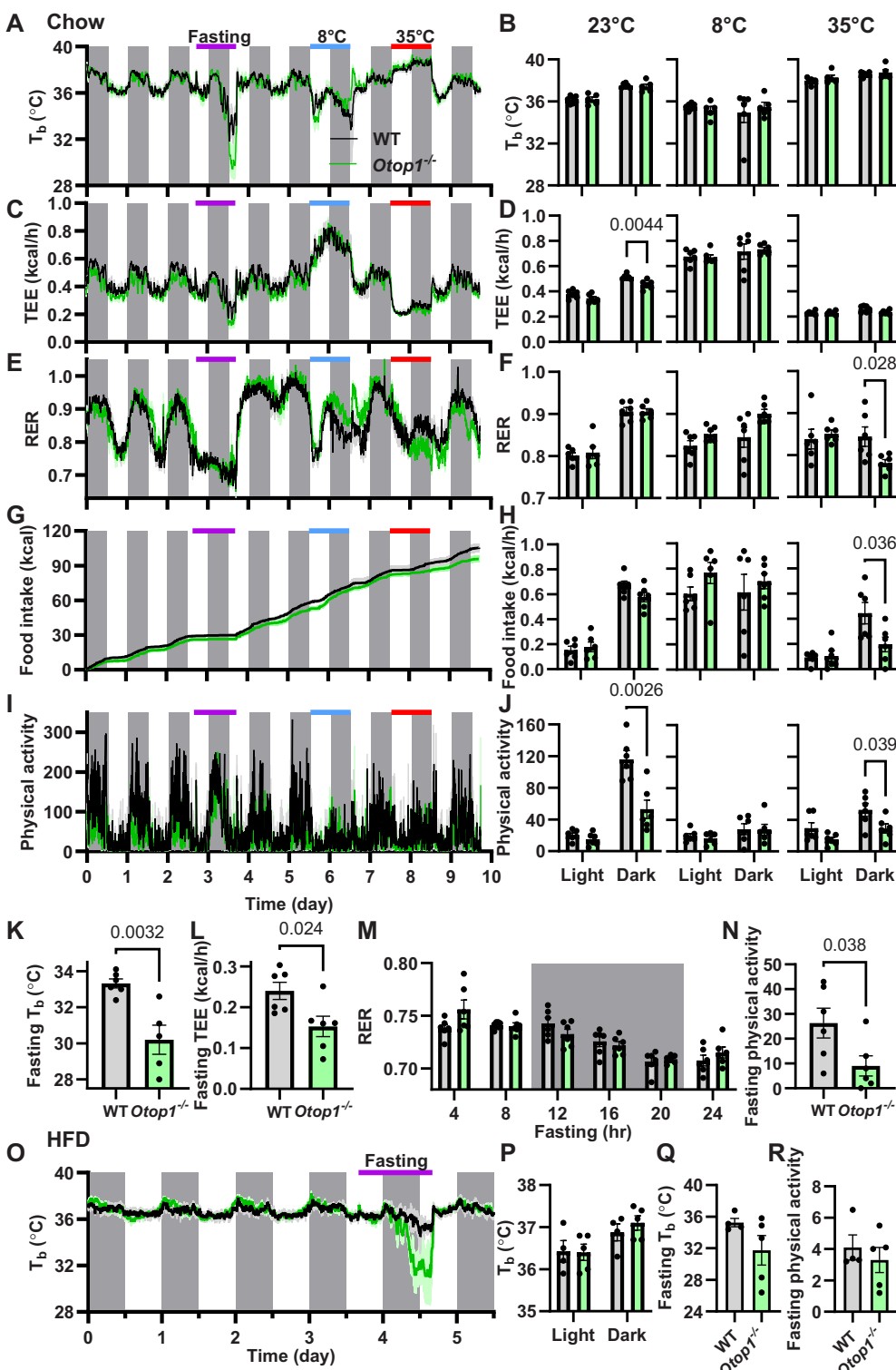

**Fig 5. *Otop1⁻/⁻* female mice also have an augmented response to fasting.** (A-J) Female 19-week-old wild type (WT) and *Otop1⁻/⁻* mice with chow diet at 23 °C were exposed to fasting (day 3), 8 °C (day 6), or 35 °C (day 8) while (A) T$_b$, (C) TEE, (E) RER (respiratory exchange ratio), (G) cumulative food intake, and (I) physical activity were measured. Mean (B) T$_b$, (D) TEE, (F) RER, (H) hourly food intake, and (J) physical activity during the light and dark periods at the indicated ambient temperature. (K) Mean T$_b$, (L) TEE, and (N) physical activity during the last 4 hours of fasting (n = 6/group, p-value calculated by unpaired t-test). (M) RER in 4-hour intervals during fasting (n = 6/group, p-value

calculated by 2-way ANOVA with Sidak post-hoc testing). (O-R) Female 36-week-old mice after 8 weeks on a high fat diet (HFD. (O) $T_b$, (P) mean baseline $T_b$, and (Q) $T_b$ and (R) physical activity during the last 4 hours of fasting (n = 4-6/group, p-values by unpaired t-test).

(MAC-2), *Cd68*, *Arg1*) in either WAT depot. Thus, the *Otop1*$^{-/-}$ mice did not demonstrate increases in markers of inflammation.

### 3.6 *Otop1*$^{-/-}$ mice have intact hypothermia after stimulation of mast cells

Since *Otop1* is highly expressed in mast cells [28], we examined mast cell function. Activation of adenosine A3 receptors (A$_3$AR) causes mast cell degranulation, histamine release, and hypothermia in mice [29]. Treatment with the A$_3$AR agonist MRS5698 produced hypothermia in both WT and *Otop1*$^{-/-}$ mice, with the *Otop1*$^{-/-}$ mice taking longer to recover (**Fig 8**). Thus, there is no evidence for reduced mast cell activation in *Otop1*$^{-/-}$ mice.

## 4 Discussion

The high level of *Otop1* expression in BAT spurred us to examine the role of OTOP1 in BAT function. The major conclusion is that OTOP1 is dispensable for basal thermoregulation. *In vivo* differences in non-basal thermal physiology between *Otop1*$^{-/-}$ and wild type mice include an amplified hypothermic response to fasting and a greater response to stimulation with a β3-adrenergic agonist, CL316243. *Ex vivo*, *Otop1*$^{-/-}$ BAT basal mitochondrial respiration was unchanged. In contrast, maximal respiration was reduced and β3-adrenergic stimulated respiration was increased when compared to controls. Further studies are required to understand the mechanistic basis underlying how the loss of OTOP1 function causes these changes.

*Otop1*$^{-/-}$ mice also show some subtle changes in $T_b$ regulation. Normally, to conserve energy during fasting mice reduce their $T_b$, ranging from a slight decrease to torpor [30]. Fasted *Otop1*$^{-/-}$ mice showed greater and/or quicker reductions in $T_b$, TEE, and RER compared to controls. The increased hypothermia/ hypometabolism does not reflect an inability to generate heat as the *Otop1*$^{-/-}$ mice show normal cold defense. A possible explanation is that the *Otop1*$^{-/-}$ mice have a reduced ability to acutely mobilize fuels, making fasting a greater metabolic challenge, thereby amplifying the hypothermia and hypometabolism. The unchanged body weight and adiposity suggest that there is no reduction in total fuel stores. The up-regulation of fatty acid transporter RNAs is consistent with a regulated response to perceived lower BAT fatty acid availability.

There are known sex differences in mouse thermal physiology, including a higher $T_b$ in females and a different response to fasting [31], with sex hormones contributing to the sex dimorphism [32]. *Otop1*$^{-/-}$ mice may have some sex differences, but additional studies are needed to better characterize these subtle findings.

Due to the high levels of *Otop1* mRNA in mast cells, we screened mast cell function in *Otop1*$^{-/-}$ mice. No deficit was detected in mast cell degranulation-induced hypothermia, although we cannot rule out that other assays might reveal a phenotype.

*Otop1*$^{tlt/tlt}$ mice on a high fat diet showed increased insulin resistance and WAT inflammation [13]. However, we did not detect an increase in insulin resistance or WAT inflammation in the current strain of *Otop1*$^{-/-}$ mice. This could be due to the different causal mutations (38-bp deletion causing frameshift vs A151E missense), with the OTOP1 tlt protein having about 10% residual activity while there was no activity from the 38-bp deletion allele [1]. Other possible explanations include different genetic backgrounds (C57BL/6J vs mostly C57BL/6J) and stochastic variation.

In summary, *Otop1*$^{-/-}$ mice have nearly normal thermoregulation, excepting exaggerated body temperature reduction during fasting and an augmented metabolic rate response to β$_3$-adrenergic stimulation. Loss of *Otop1* also alters stimulated BAT mitochondrial function in a

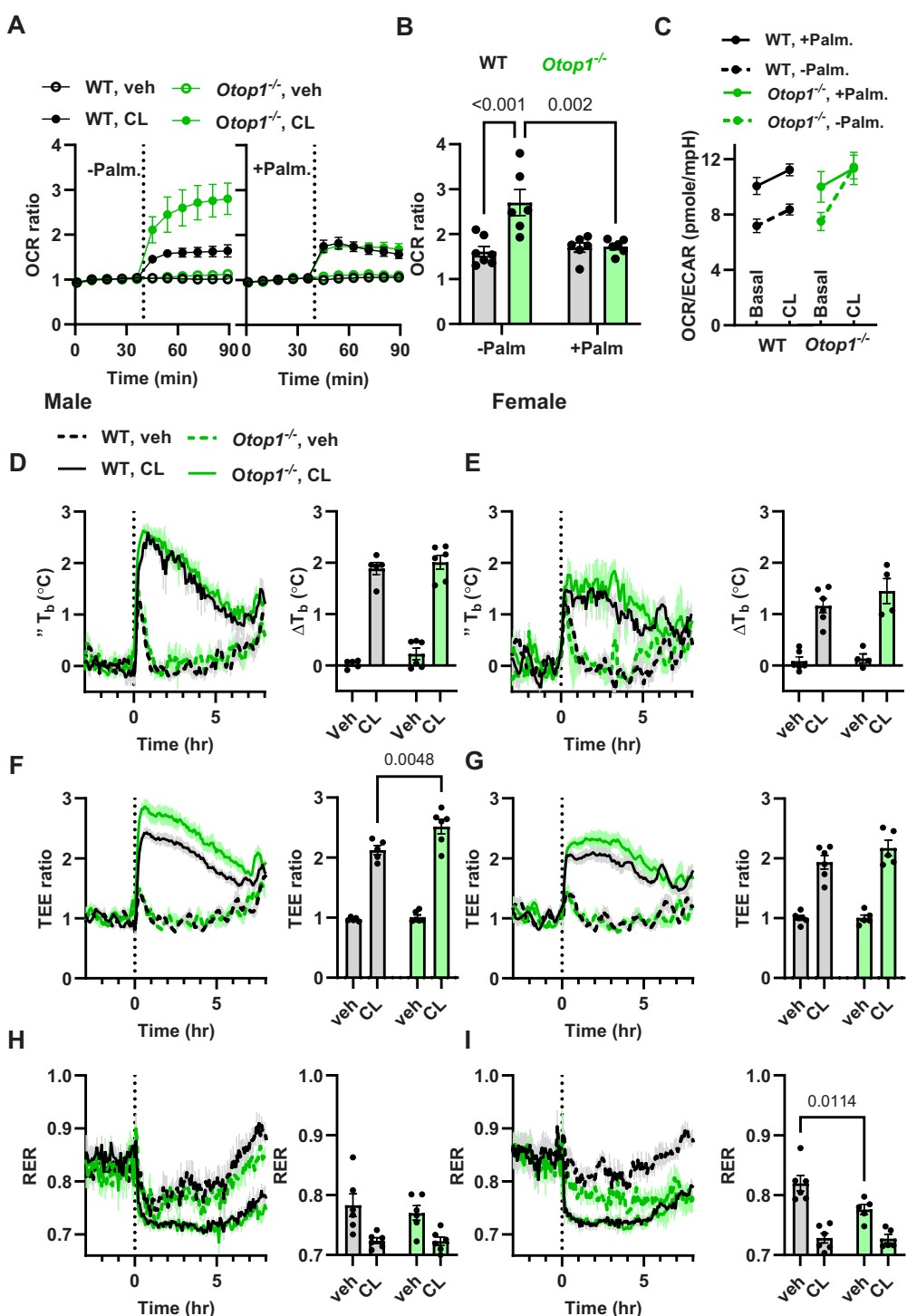

**Fig 6. Loss of Otop1 increases CL316243-induced energy expenditure.** (A) *Ex vivo* effect of CL316243 (10 μM) treatment on BAT oxygen consumption rate (OCR), without or with added palmitate. (B) Ratio of CL316243-treated to basal OCR. (C) OCR:ECAR ratio before and after CL316243 stimulation (mean ± s.e.m, 10- to 12-week-old mice; p-values from 2-way ANOVA with Tukey post-hoc testing). (D-I) *In vivo* treatment with CL316243 (0.1 mg/kg, ip, at time 0) or vehicle at 30 °C in 14-week-old male and female mice (D, E) Body temperature change ($\Delta T_b$), (F, G) total energy expenditure (TEE) ratio, and (H, I) RER. $\Delta T_b$ is the change from baseline (mean of 0–5 hours minus mean of -2.5 to -0.5 hours). TEE ratio is the ratio to baseline using the same intervals, n = 5-6/group, p-values from 2-way ANOVA with Sidak post-hoc testing.

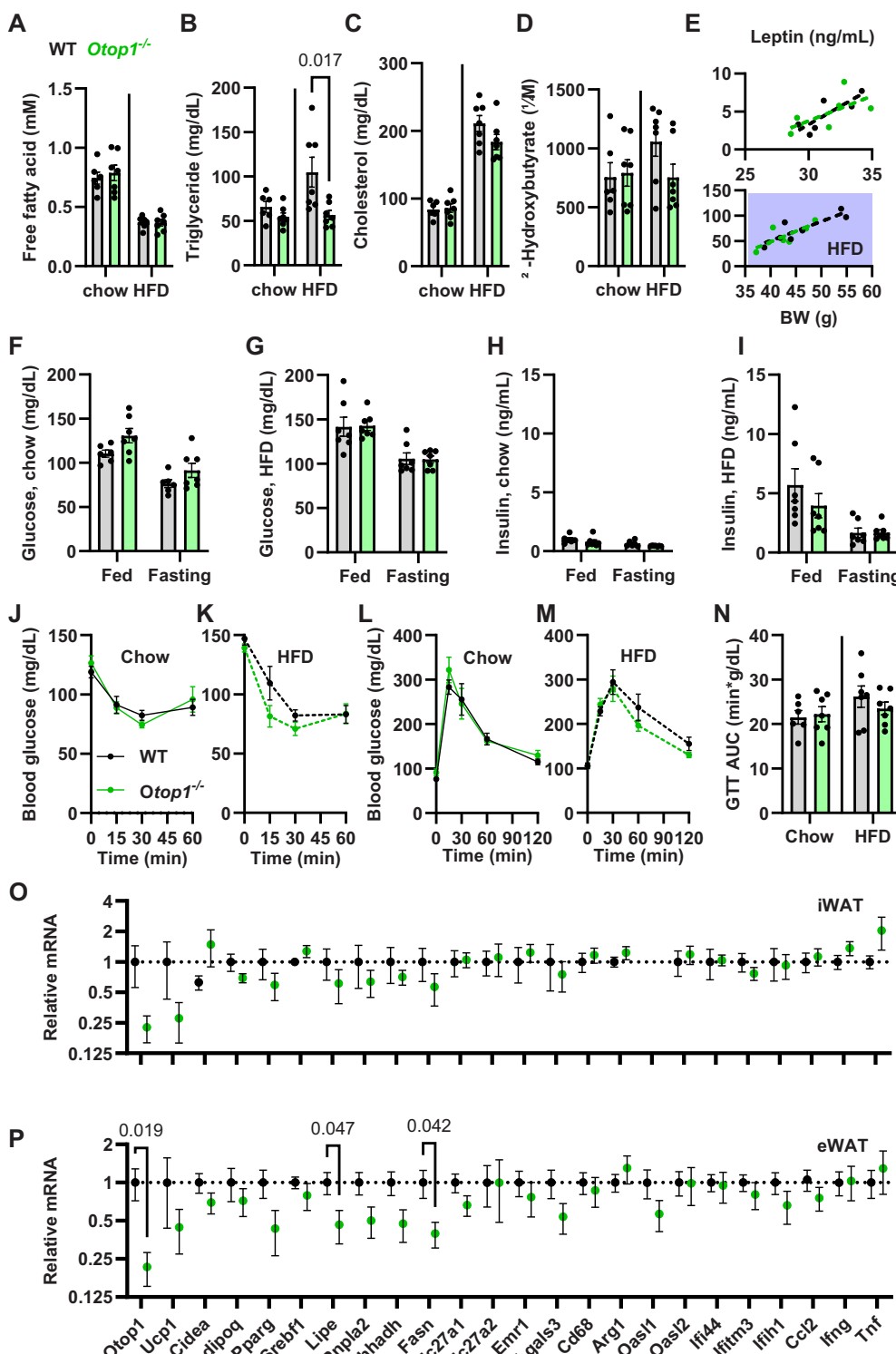

**Fig 7. Metabolic parameters in *Otop1⁻/⁻* male mice.** (A-I) Plasma metabolite levels in littermate male chow- and HFD-fed mice. (J,K) Insulin tolerance test (0.75 u/kg) in chow and HFD mice. (L-N) Glucose tolerance test in chow (2 g/kg) and HFD (1 g/kg) mice. Chow mice were studied at ~25 weeks (mean body weights: WT, 30.7 g; *Otop1⁻/⁻* 30.4 g). HFD mice were studied at ~37 weeks, after 8 weeks on the HFD diet (mean body weights: WT, 46.1 g; *Otop1⁻/⁻* 42.8 g). n = 6-7/group, p-values from unpaired t-test. (O,P) mRNA levels in inguinal and epididymal WAT from littermate mice on an HFD. mRNA was normalized to *Tbp* and expressed relative to WT (n = 6/group, p-values from unpaired t-test).

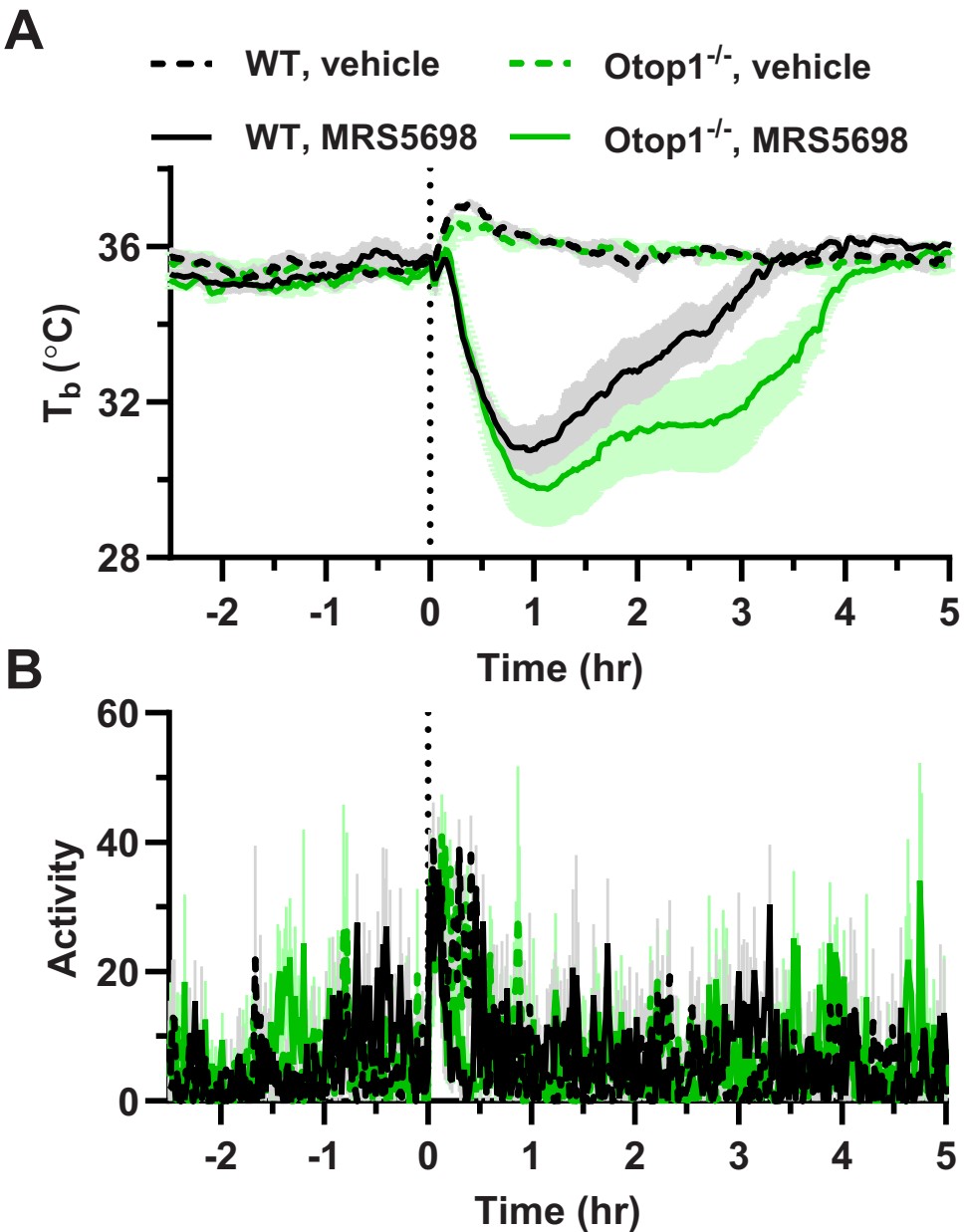

**Fig 8. Effect of adenosine A₃AR agonist in *Otop1*<sup>-/-</sup> mice.** (A) Body temperature (T_b) and (B) physical activity in male 29-week-old WT (black) and *Otop1*<sup>-/-</sup> (green) mice treated with MRS5698 (10 mg/kg, i.p.) or vehicle at time 0. Examining 1 h intervals, Tb in MRS5698-treated *Otop1*<sup>-/-</sup> was lower than WT from hour 3 to 4 ($p = 0.021$ by unpaired t-test) and hour 4 to 5 ($p = 0.006$). Activity was not significantly different between *Otop1*<sup>-/-</sup> and WT.

substrate-dependent manner. Further investigation is required to understand the mechanistic basis of these observations.

## Supporting information

**S1 Fig. RNA-Seq shows a 38bp deletion in exon 1 of Otop1 in BAT of *Otop1*<sup>-/-</sup> mice.** BAT mRNA from 3 male mice of each genotype was studied. Mapped reads from RNA-Seq in exon 1 of the *Otop1* gene, with the site of the 38-bp deletion indicated with a box. Black: WT; Red:

*Otop1^-/-*. Each row is a different mouse.
(PDF)

**S2 Fig. Gene expression of BAT mRNA by qPCR.** (A) Volcano plot of BAT RNA-Seq results. Gene names are indicated for the 10 most significantly different RNAs between wild type and *Otop1^-/-* mice. BAT mRNA level by qPCR in chow-fed WT and *Otop1^-/-* male mice (aged 17–19 weeks) for (B) differentially expressed genes identified by RNA-Seq and (C) known BAT genes. n = 6/group; p-value from unpaired t-test. (D) BAT mRNA level by qPCR in HFD-fed WT and *Otop1^-/-* male mice (aged 45 weeks) for known BAT genes. n = 6/group; p-value from unpaired t-test. mRNA levels were normalized to *Tbp* gene.
(PDF)

**S3 Fig. Gene expression in BAT from WT and *Otop1^-/-* mice after a 24-hour fast.** BAT mRNA from male mice measured by qPCR (Black: WT; Green; *Otop1-/-*; n = 6–7; p-value calculated by t-test).
(PDF)

**S4 Fig. Loss of *Ucp1* decreases basal oxygen consumption rate (OCR) and response to CL316243 in brown adipose tissue (BAT).** (A) OCR ratio (normalized to basal OCR) in response to CL316243. (B) CL:Basal OCR ratio is the mean of 6 measurements after CL316243 application. n = 6/group; p-value from unpaired t-test. (C) OCR/ECAR ratio. n = 6/group.
(PDF)

**S5 Fig. Loss of *Ucp1* decreases body temperature and energy expenditure stimulated by β3-agonist.** (A) Body temperature (Tb), (B) total energy expenditure (TEE), and (C) RER of male mice from the indicated genotypes, in response to CL316243 (0.1 mg/kg, ip., injected at time 0). The bar graphs show the mean at 0 to 5 hr. n = 5-6/group; p-values from 3-way ANOVA with Tukey post-hoc testing.
(PDF)

**S6 Fig. Metabolic parameters in *Otop1^-/-* female mice.** (A-I) Plasma metabolite levels in littermate male chow- and HFD-fed mice. (J,K) Insulin tolerance test (0.75 u/kg) in chow and HFD mice. (L-N) Glucose tolerance test in chow (2 g/kg) and HFD (1 g/kg) mice. Chow mice were studied at ~23 weeks (mean body weights: WT, 24.5 g; *Otop1^-/-* 23.3 g). HFD mice were studied at ~34 weeks, after 8 weeks on the HFD diet (mean body weights: WT, 35.1 g; *Otop1^-/-* 30.2 g; p = 0.13). n = 4-7/group, p-values from unpaired t-test.
(PDF)

**S7 Fig. Original blot for Fig 1I.**
(PPTX)

**S1 Table. PCR primer sets.**
(PDF)

**S2 Table. RNA-seq data by mouse.**
(XLSX)

## Acknowledgments

We thank Alice Franks and Yinyan Ma for superb assistance.

## Author Contributions

**Conceptualization:** Yu-Hsiang Tu, Oksana Gavrilova, Marc L. Reitman.

**Data curation:** Yu-Hsiang Tu.

**Funding acquisition:** Oksana Gavrilova, Marc L. Reitman.

**Investigation:** Yu-Hsiang Tu, Naili Liu, Cuiying Xiao, Oksana Gavrilova.

**Methodology:** Yu-Hsiang Tu.

**Writing – original draft:** Yu-Hsiang Tu, Marc L. Reitman.

**Writing – review & editing:** Yu-Hsiang Tu, Naili Liu, Cuiying Xiao, Oksana Gavrilova, Marc L. Reitman.

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
