## [Decision Letter · Decision Letter 0]

18 Jan 2023

PONE-D-22-34443Physiology of mice lacking Otopetrin 1 (OTOP1) functionPLOS ONE

Dear Dr. Reitman,

Thank you for submitting your manuscript to PLOS ONE. After careful consideration, we feel that it has merit but does not fully meet PLOS ONE’s publication criteria as it currently stands. Therefore, we invite you to submit a revised version of the manuscript that addresses the points raised during the review process.

We look forward to receiving your revised manuscript.

Kind regards,

Aijun Qiao, Ph.D.

Academic Editor

PLOS ONE

“We thank Alice Franks and Yinyan Ma for superb assistance. This research was supported by the Intramural Research Program of the National Institutes of Health, National Institute of Diabetes and Digestive and Kidney Diseases (ZIA DK075062, ZIA DK075064, ZIA DK070002).”

“This research was supported by the Intramural Research Program of the National Institutes of Health, National Institute of Diabetes and Digestive and Kidney Diseases (MLR, ZIA DK075062; MLR, ZIA DK075064;OG ZIA DK070002). The funders had no role in study design, data collection and analysis, decision to publish, or preparation of the manuscript.”

Reviewers' comments:

Reviewer's Responses to Questions

**Comments to the Author**

1. Is the manuscript technically sound, and do the data support the conclusions?

Reviewer #1: No

Reviewer #2: Partly

Reviewer #3: Partly

2. Has the statistical analysis been performed appropriately and rigorously? 

Reviewer #1: I Don't Know

Reviewer #2: Yes

Reviewer #3: Yes

3. Have the authors made all data underlying the findings in their manuscript fully available?

Reviewer #1: Yes

Reviewer #2: Yes

Reviewer #3: Yes

4. Is the manuscript presented in an intelligible fashion and written in standard English?

Reviewer #1: Yes

Reviewer #2: Yes

Reviewer #3: Yes

5. Review Comments to the Author

Reviewer #1: OTOP1 is a proton channel that is highly expressed in BAT. In the present study, Tu et al. found that Otop1-/- mice had similar body temperature as control mice at baseline and in response to cold and hot temperature. Interestingly, Otop1-/- mice exhibited an exaggerated hypothermia in response to fasting. Oxygen consumption upon stimulation with CL316243 (b3-adrenergic agonist) was significantly increased. A previous study reported that OTOP1 mutant mice exhibit insulin resistance, but the insulin sensitivity was normal in this study. The authors concluded that loss of OTOP1 does not change basal function of BAT, but affects stimulated responses. The findings are potentially interesting, but there are several questions that deflate the value of the present study.

Major criticisms

Figure 1. Western blotting of OTOP1 is necessary.

Figure 2A and B. The authors described that “Otop1-/- male and female mice had the same body weights as controls”. However, it looks (especially female) that there is an attenuation of weight gain in the mutant mice. The authors started HFD feeding around 26-30 weeks of age. It is unusual (too late).

Figure 3. The authors should show the UCP1 protein expression levels in this context. Palmitate affects OCR of the mutant mice. Why?

Figure 4. The body temperature of Otop1-/- mice in the fasting condition was significantly lower. Diet-induced thermogenesis may be impaired in the mutant mice. Decrease of physical activity at 23C might be also an important reason. What is the mechanism of reduced physical activity? How about the body temperature and physical activity of BAT specific Otop1 KO mice? More experiments and explanation are necessary. The decrease of body temperature was not detected in the mutant mice fed a HFD. Why?

Figure 6. UCP1 plays an important role in the b3 agonist-induced oxygen consumption. Western blotting of UCP1 is required.

Figure 7. They indicate that insulin sensitivity of the mutant mice is similar. This reviewer thinks that the sensitivity is better in the mutant (Figure 7K).

Supplementary Figure 6 and discussion. The authors claimed that FCCP induced intracellular acidification and CL316243 (b3 agonist) induced intracellular alkalization. They propose that changes of pH cause protein influx/efflux. However, there are no direct experimental evidence to support the proposal.

Reviewer #2: This manuscript examined the thermogenic phenotype of Otopetrin 1 (Otop1)-deficient mice focusing on dissecting the role of Otop1 in brown adipose tissue. In general, systemic characterization of energy homeostasis by indirect calorimetry identified maintained body temperature at baseline and in response to cold and hot ambient temperature in Otop1-deficient mice, suggesting an indispensable role of Otop1 in BAT function despite its high expression. They did identify that Otop1-/- mice exhibited exaggerated hypothermia and hypometabolism under fasting without providing any explanation. They further analyzed BAT mitochondrial function by Seahorse analysis in the presence or absence of palmitate or after stimulation of CL316243. These data were rather confusing and failed to provide a clear picture of how exactly Otop1 loss exaggerates CL316243 stimulated response in BAT. The following concerns need to be addressed:

The manuscript title needs to be more specific: e.g. Otopetrin 1 is dispensable for thermoregulation in mice.

Why there is only a 35% reduction of Otop1 mRNA in Otop1-/- mice? Western blot is needed to confirm the deletion of Otop1 in BAT, if possible other tissues in global Otop1-/- mice. Whether the low efficiency of Otop1 deletion explains the lack of similar metabolic phenotypes observed in HFD-fed Otop1tlt/tlt mice (ref 15). The discrepancy between these two mouse lines needs to be better discussed.

How the seahorse analysis of BAT mitochondrial function was exactly measured is confusing. The methodology mentioned collagenase digestion of BAT and filtered retained tissues were used for seahorse analysis. This is problematic, as no data showed that the retained tissue is the BAT. Collagenase digestion is expected to fractionate primary brown adipocytes from the Stromal vascular cell fraction. The retained tissues could be simply undigested fibrotic tissues. The cited reference (line 152, Refs 20 & 21) clearly used BAT explant without collagenase digestion. Meanwhile, it is also puzzling why Palmitate did not increase basal respiration as compared to vehicle-treated one, since FFA is expected to activate FAO and uncoupling, increasing respiration. In addition, mitostress test is not a good way to assess glycolytic capacity. It is better to be measured by glycolysis stress test. Therefore, Figure 3 and 6 are questionable and needs to be clarified.

The energy balance in males at 23°C does not add up together as these male Otop1-/- mice have higher food intake, lower physical activity, and higher RER without changes in TEE and body weights. Please explain.

What explains the sex differences in thermogenesis of Otop1-/- mice at different temperatures and after CL injection needs to be discussed.

I’m not sure why RNA-seq data was only selectively presented in Fig. 8. How many overall DEGs were identified? Are there any pathways enriched? This information may give an overview of the physiological changes of Otop1-/- BAT. Furthermore, these data need to be confirmed by RT-PCR in BAT.

Not sure how similar insulin sensitivity resulted in improved glucose tolerance in HFD-fed Otop1-/- mice (Supplemental Figure 4K&M).

Insulin tolerance test is normally performed after a short (4-6h) fast instead of under nonfasting conditions.

Discussion on Supplemental Figure 6 is too much speculation. Current data provides no such insights.

Please indicate at what age the mice were started on HFD in Materials and Methods.

Please indicate what software was used to perform statistic analyses.

Reviewer #3: The manuscript by Tu et al. reported the function of OTOP1 in BAT, where OTOP1 expression level was enriched, using OTOP1 deficient mice. Although HFD-induced body weight gain was comparable, fasting-induced energy expenditure was reduced in OTOP1-deficient mice. Also, the oxygen consumption rate in BAT from OTOP1 KO mice was diminished upon FCCP treatment, whereas it was enhanced upon CL treatment.

Unfortunately, the purpose and interpretation of each figure are unclear. Thus, it is hard to understand the physiological roles of OTOP1. Moreover, provided data are not sufficient to suggest BAT is the main organ mediating those metabolic changes in OTOP1-deficient mice. Furthermore, it is likely that the authors should collect more data showing that OTOP1-mediated metabolic change is mediated through its proton channel function. Followings are specific comments to improve the study.

1. OTOP1 BAT showed different results on OCR, which was reduced upon FCCP (Figure 3A) but elevated upon CL treatment (Figure 6A). Also, palmitate addition showed different effects on the OCR in each experiment. The authors described that these results could be associated with the influx or efflux of protons depending on cellular pH (text lines 265-279). However, the authors did not provid the data showing proton flux. Moreover, it is possible that palmitate could act as an energy source or lipotoxic mediator rather than a proton carrier. Thus, for the author’s proposed model, they need show intracellular pH or proton flux to elucidate the roles of OTOP1 and palmitate in those experiments.

2. In Figure 3, the authors described that ‘these data show that Otop1 loss reduces maximal, but not basal respiration, in contrast to the effect of Ucp1 loss which reduces basal respiration’ (text lines 171-174). However, basal respiration cannot be compared because WT is not included in Supplementary Figure 2. Thus, it needs to show the OCR of BATs from all four groups (WT vs. OTOP1 KO vs. UCP1 KO vs. OTOP1/UCP1 DKO) in one graph to compare basal and FCCP-induced OCR.

3. Related to Figures 4 and 5, it needs to explain the rationale for the author’s hypothesis (text p.5, lines 178-180) regarding the thermal physiology experiments. Also, the authors should describe the aims of this experimental design (23 ℃ -> fasting -> 8 ℃ -> 35 ℃).

4. One of the consistent changes in metabolic phenotypes is the reduced Tb and TEE in OTOP1 KO mice (Figures 4K-L, Figures 5K-L). The authors should investigate the roles of BAT OTOP1 in the fasting condition. The comparison of histology or thermogenic gene expression in BATs from control and OTOP1 KO mice upon fasting might be informative to reveal the function of BAT OTOP1-mediated thermogenesis or energy expenditure.

5. In Figure 6, the authors have to show the CL responses (lipid droplet size by histology and metabolic gene expression levels, including thermogenic genes) in BATs from WT and OTOP1 KO mice to examine the roles of BAT OTOP1 in CL-mediated energy expenditure

6. PLOS authors have the option to publish the peer review history of their article (what does this mean?). If published, this will include your full peer review and any attached files.

Reviewer #1: No

Reviewer #2: **Yes: **Weiqin Chen

Reviewer #3: No

---

## [Author Response · Author response to Decision Letter 0]

1 May 2023

please see attached 'response to reviewers' file

---

## [Decision Letter · Decision Letter 1]

29 Jun 2023

PONE-D-22-34443R1Loss of Otopetrin 1 affects thermoregulation during fasting in micePLOS ONE

Dear Dr. Reitman,

Thank you for submitting your manuscript to PLOS ONE. After re-evaluated by prvious reviewers and careful consideration by board members, we feel that it has merit but remains not fully meet PLOS ONE's publication criteria as it currently stands. Therefore, we invite you to submit a revised version of the manuscript that carefully addresses the points raised by reviewer 3 (see below) during the review process. "The authors have made efforts to address the reviewer’s comments in revised manuscript. However, the revised manuscript does not clearly demonstrate this reviewer’s suggestions. In particular, the authors describe the role of OTOP1 in fasting-induced hypothermia and increased CL-mediated thermogenesis in OTOP1 KO mice in the discussion section (lines 291-305), but there is a lack of experimental evidence supporting their proposed model. Thus, it is difficult to accept the conclusions without any experimental evidence. To strengthen the proposed model, the authors have to compare the intracellular pH between WT and OTOP1 deficient brown adipocyte in response to FCCP and beta3 adrenergic agonist. This can be accomplished by utilizing appropriate dyes or a pH-sensitive indicator such as pHluorin (pH-sensitive indicator pHluorin, reference DOI: https://doi.org/10.7554/eLife.77946). This data would provide direct evidence for the involvement of OTOP1 in modulating intracellular pH under fasting or CL conditions."

We look forward to receiving your revised manuscript.

Kind regards,

Aijun Qiao, Ph.D.

Academic Editor

PLOS ONE

Journal Requirements:

Reviewers' comments:

Reviewer's Responses to Questions

**Comments to the Author**

1. If the authors have adequately addressed your comments raised in a previous round of review and you feel that this manuscript is now acceptable for publication, you may indicate that here to bypass the “Comments to the Author” section, enter your conflict of interest statement in the “Confidential to Editor” section, and submit your "Accept" recommendation.

Reviewer #1: All comments have been addressed

Reviewer #2: All comments have been addressed

Reviewer #3: (No Response)

2. Is the manuscript technically sound, and do the data support the conclusions?

Reviewer #1: Yes

Reviewer #2: Yes

Reviewer #3: Partly

3. Has the statistical analysis been performed appropriately and rigorously? 

Reviewer #1: Yes

Reviewer #2: Yes

Reviewer #3: Yes

4. Have the authors made all data underlying the findings in their manuscript fully available?

Reviewer #1: Yes

Reviewer #2: Yes

Reviewer #3: Yes

5. Is the manuscript presented in an intelligible fashion and written in standard English?

Reviewer #1: Yes

Reviewer #2: Yes

Reviewer #3: Yes

6. Review Comments to the Author

Reviewer #1: The authors revised the manuscript properly according to my comments. I have no conflict of interests.

Reviewer #2: (No Response)

Reviewer #3: The authors have made efforts to address the reviewer’s comments in revised manuscript. However, the revised manuscript does not clearly demonstrate this reviewer’s suggestions. In particular, the authors describe the role of OTOP1 in fasting-induced hypothermia and increased CL-mediated thermogenesis in OTOP1 KO mice in the discussion section (lines 291-305), but there is a lack of experimental evidence supporting their proposed model. Thus, it is difficult to accept the conclusions without any experimental evidence. To strengthen the proposed model, the authors have to compare the intracellular pH between WT and OTOP1 deficient brown adipocyte in response to FCCP and beta3 adrenergic agonist. This can be accomplished by utilizing appropriate dyes or a pH-sensitive indicator such as pHluorin (pH-sensitive indicator pHluorin, reference DOI: https://doi.org/10.7554/eLife.77946). This data would provide direct evidence for the involvement of OTOP1 in modulating intracellular pH under fasting or CL conditions.

7. PLOS authors have the option to publish the peer review history of their article (what does this mean?). If published, this will include your full peer review and any attached files.

Reviewer #1: No

Reviewer #2: **Yes: **Weiqin Chen

Reviewer #3: No

---

## [Author Response · Author response to Decision Letter 1]

28 Aug 2023

Response to Reviewers:

Reviewer #3: The authors have made efforts to address the reviewer’s comments in revised manuscript. However, the revised manuscript does not clearly demonstrate this reviewer’s suggestions. In particular, the authors describe the role of OTOP1 in fasting-induced hypothermia and increased CL-mediated thermogenesis in OTOP1 KO mice in the discussion section (lines 291-305), but there is a lack of experimental evidence supporting their proposed model. Thus, it is difficult to accept the conclusions without any experimental evidence. To strengthen the proposed model, the authors have to compare the intracellular pH between WT and OTOP1 deficient brown adipocyte in response to FCCP and beta3 adrenergic agonist. This can be accomplished by utilizing appropriate dyes or a pH-sensitive indicator such as pHluorin (pH-sensitive indicator pHluorin, reference DOI: https://doi.org/10.7554/eLife.77946). This data would provide direct evidence for the involvement of OTOP1 in modulating intracellular pH under fasting or CL conditions.

The reviewer requested experimental support for the model proposed in the Discussion of the revised manuscript. As suggested by the reviewer, we set up a fluorescence assay to measure intracellular pH using the dye BCECF. This work is new to our lab and, to summarize many experiments, we have not been able to reproduce the expected alkalinization of intracellular pH upon adrenergic stimulation (doi: 10.1152/ajpcell.1994.267.2.C349) with CL316243 and acidification with FCCP treatment (doi: 10.1152/ajpheart.00932.2012). Thus, to satisfy the reviewer’s suggestion, we have completely removed the model from the discussion—lines 291-305 become lines 291-294. We believe that these changes satisfy the concerns of the reviewer and editor. 

Original:

“Ex vivo, Otop1-/- BAT basal mitochondrial respiration was unchanged, while maximal respiration was reduced and β3-adrenergic stimulated respiration was increased when compared to controls. Basally, taking into account the intracellular pH, extracellular pH, and plasma membrane potential, there is a small electrochemical gradient across the brown adipocyte plasma membrane that predicts inward flux of protons via OTOP1 [30, 31]. In the presence of FCCP, the plasma membrane is depolarized and the cytosol is acidified [32]; under these conditions OTOP1 could instead mediate proton efflux, thereby increasing intracellular pH. With FCCP, the lack of OTOP1 function might decrease the intracellular pH, potentially inhibiting processes including CPT1-mediated fatty acid transport [33] and glycolysis [34], thereby leading to the observed reduction in Otop1-/- OCR. 

Under stimulated conditions, such as with a β3-adrenergic agonist, BAT cytosol is alkalinized [31]. Cytoplasmic alkalinization could increase OTOP1-mediated proton influx across the plasma membrane. Thus, loss of Otop1 would reduce that pH gradient across the inner mitochondrial membrane and thus cause a compensatory increase mitochondrial respiration to maintain the pH gradient. Additional palmitate could compensate for the loss of Otop1, as its protonated form might act as carrier to increase proton influx.”

Revised:

“Ex vivo, Otop1-/- BAT basal mitochondrial respiration was unchanged. In contrast, maximal respiration was reduced and β3-adrenergic stimulated respiration was increased when compared to controls. Further studies are required to understand the mechanistic basis underlying how the loss of OTOP1 function causes these changes.”

---

## [Decision Letter · Decision Letter 2]

26 Sep 2023

Loss of Otopetrin 1 affects thermoregulation during fasting in mice

PONE-D-22-34443R2

Dear Dr. Reitman

We are pleased to inform you that your manuscript has been judged scientifically suitable for publication and will be formally accepted for publication once it meets all outstanding technical requirements.

Kind regards,

Aijun Qiao, Ph.D.

Academic Editor

PLOS ONE

Additional Editor Comments (optional):

Reviewers' comments:

Reviewer's Responses to Questions

**Comments to the Author**

1. If the authors have adequately addressed your comments raised in a previous round of review and you feel that this manuscript is now acceptable for publication, you may indicate that here to bypass the “Comments to the Author” section, enter your conflict of interest statement in the “Confidential to Editor” section, and submit your "Accept" recommendation.

Reviewer #3: All comments have been addressed

2. Is the manuscript technically sound, and do the data support the conclusions?

Reviewer #3: Yes

3. Has the statistical analysis been performed appropriately and rigorously? 

Reviewer #3: Yes

4. Have the authors made all data underlying the findings in their manuscript fully available?

Reviewer #3: Yes

5. Is the manuscript presented in an intelligible fashion and written in standard English?

Reviewer #3: Yes

6. Review Comments to the Author

Reviewer #3: The authors sincerely tried to conduct experiments strictly following the reviewer's comments. Their meticulous attention to detail thoroughly addresses most of the reviewer's suggestions and concerns. As a result, the revised manuscript can be considered acceptable and significantly improved.

7. PLOS authors have the option to publish the peer review history of their article (what does this mean?). If published, this will include your full peer review and any attached files.

Reviewer #3: No

---

## [Editor Report · Acceptance letter]

28 Sep 2023

PONE-D-22-34443R2 

Loss of Otopetrin 1 affects thermoregulation during fasting in mice 

Dear Dr. Reitman:

I'm pleased to inform you that your manuscript has been deemed suitable for publication in PLOS ONE. Congratulations! Your manuscript is now with our production department. 

Kind regards, 

on behalf of

Dr. Aijun Qiao 

Academic Editor

PLOS ONE